# Treatment with Pulsed Extremely Low Frequency Electromagnetic Field (PELF-EMF) Exhibit Anti-Inflammatory and Neuroprotective Effect in Compression Spinal Cord Injury Model

**DOI:** 10.3390/biomedicines10020325

**Published:** 2022-01-29

**Authors:** Yona Goldshmit, Moshe Shalom, Angela Ruban

**Affiliations:** 1Steyer School of Health Professions, Sackler Faculty of Medicine, Tel-Aviv University, Tel Aviv 6997801, Israel; yona.goldshmit@tauex.tau.ac.il (Y.G.); mosheshalom@mail.tau.ac.il (M.S.); 2Australian Regenerative Medicine Institute, Monash Biotechnology, 15 Innovation Walk, Melbourne 3800, Australia; 3Sagol School of Neuroscience, Tel Aviv University, Tel Aviv 6997801, Israel

**Keywords:** spinal cord injury, electromagnetic field, neuroprotection, treatment, rehabilitation

## Abstract

Background: Spinal cord injury (SCI) pathology includes both primary and secondary events. The primary injury includes the original traumatic event, and the secondary injury, beginning immediately after the initial injury, involves progressive neuroinflammation, neuronal excitotoxicity, gliosis, and degeneration. Currently, there is no effective neuroprotective treatment for SCI. However, an accumulating body of data suggests that PELF-EMF has beneficial therapeutic effects on neurotrauma. The purpose of this study was to test the efficacy of the PELF-EMF SEQEX device using a compression SCI mouse model. Methods: C57BL/6 mice were exposed to PELF-EMF for 4 h on a daily basis for two months, beginning 2 h after a mild-moderate compression SCI. Results: The PELF-EMF treatment significantly diminished inflammatory cell infiltration and astrocyte activation by reducing Iba1, F4/80, CD68+ cells, and GAFP at the lesion borders, and increased pro-survival signaling, such as BDNF, on the neuronal cells. Moreover, the treatment exhibited a neuroprotective effect by reducing the demyelination of the axons of the white matter at the lesion’s center. Conclusions: Treatment with SEQEX demonstrated significant anti-inflammatory and neuroprotective effects. Considering our results, this safe and effective rehabilitative device, already available on the market, may provide a major therapeutic asset in the treatment of SCI.

## 1. Introduction

Spinal cord injury (SCI) is a traumatic event that results in altered sensory and motor function that can ultimately affect a patient’s physical, psychological, and social well-being [1]. Statistics show that more than 27 million people worldwide suffer from long-term disabilities due to spinal cord injuries, 90% of which are due to trauma [2]. The life expectancy of SCI patients highly depends on the level of injury and the degree of preserved functions. Today, the estimated lifetime cost of a SCI patient is $2.35 million [3]. Although there are extensive efforts underway to develop novel treatments for SCI, the current clinical treatment remains far from ideal [4,5]. Therapy for SCI is not adequately effective and consists mostly of hemodynamic stabilization, spinal cord decompression, and early rehabilitation [6]. Patient outcome is largely determined by the management of resultant symptoms and rehabilitation to maximize neural function [7]; however, rehabilitation is a long process that requires full patient cooperation [8,9]. It is therefore critical to explore and develop new rehabilitative interventions to improve the patients’ neurological and functional outcomes.

The pathophysiology of SCI is multilayered, including primary and secondary events. The primary injury results from traumatic mechanical force, producing diffuse or focal pathology. This initial phase is characterized by tissue deformation, membrane depolarization, disruption of blood vessels and axons, ischemia, and cell membrane damage [10,11]. The secondary injury begins within minutes of the initial injury and continues for months to years [7]. This secondary injury is triggered in response to the primary damage and leads to progressive neuroinflammation, neuronal excitotoxicity, axonal demyelination and degeneration, and ultimately, neuronal cell death [11,12,13,14]. While therapeutic efforts are often focused on the axon regeneration of the central nervous system (CNS), astrocytic and glial secondary responses significantly potentiate SCI damage [14,15]. During the first minutes following injury, microglial activation and monocyte/macrophage infiltration initiates into the CNS, and their numbers remain high for several months following injury [11,16,17]. Activated microglia/macrophages clear tissue debris and are part of the body’s response to trauma [18]. Their prolonged activation, however, contributes to the secondary damage, or to the CNS injury, by releasing pro-inflammatory cytokines, neurotoxic factors, and reactive oxygen species that induce neurodegeneration at the injury site [19,20]. In addition to this, astrocytes react acutely to the CNS injury by increasing cytokine and chemokine production [21]. The results from animal studies suggest that limiting pro-inflammatory damage can improve SCI outcomes [22,23,24].

Few studies have examined the effects of using extremely low frequency electromagnetic field (ELF-EMF) in SCI and stroke models, but its use has demonstrated decreased neuronal/axonal degeneration and scar formation, increased neuronal plasticity, and improved functional recovery in animal models [25,26,27]. Exposing rats with SCI to PELF-EMF stimulation (F = 50 Hz, Mf = 1 mT) for 4 h a day decreased the levels of TNF-α and IL-6, promoted remyelination, and increased the expression of BDNF [25]. In an ischemic stroke animal model, ELF-EMF treatment (F= 15.72 Hz, Mf = 10 μT) two days after occlusion increased neuronal regeneration and the expression of nestin and doublecortin (DCX) [26]. In a similar study, treatment (F = 60 Hz, Mf = 10 mT) started within 30–40 min post occlusion decreased IL-1β and MMP9 levels and modulated the apoptotic cascade by promoting activation of the BDNF/TrkB/Akt signaling pathway, thus decreasing the expression of caspase-3 [27]. Furthermore, it has been reported that ELF-EMF exposure induces the early expression of neuronal differentiation markers, likely due to the opening of the L-type voltage-gated calcium channels via increased Ca^2+^ ions [28]. It has also been demonstrated that low frequency electromagnetic pulses reduced the IL-1β and TNF-α in nucleus pulposus cells and downregulate TNF-α as well as its transcription factor, nuclear factor kappa B (NFkB), in murine macrophages [29]. ELF-EMF-induced membrane depolarization has been shown to promote neuronal survival [30], probably through the above mechanisms. 

The knowledge about the importance of early rehabilitation and the current lack of effective neuroprotective therapy brings us to examine an available and simple-to-use PELF-EMF device as a potential rehabilitative treatment in the SCI mouse model for mild to moderate compression injuries.

## 2. Materials and Methods

### 2.1. Mice

Male and female C57BL/6 adult mice (two months) were used in this study. We used young adult mice, as about 50% of spinal cord injuries occur in young people between the ages of 16 and 30 years. All experiments were conducted according to the Guidelines for the Use of Experimental Animals of the European Community and were approved by the Animal Care Committee of Tel Aviv University.

### 2.2. Spinal Cord Compression

All mice (25–35 g) were anesthetized by intraperitoneal injections of ketamine (60 mg/kg) and xylazine (10 mg/kg). The skin was shaved and wiped with 70% ethanol prior to making a longitudinal cut through the dorsal surface. Muscles and the laminal arches of thoracic vertebrae T8-T12 were removed, and the spinal cord was exposed. Calibrated Dumont #5 forceps (with a spacer of 0.3 mm) were placed in approximately the middle of the exposed segment of the spinal cord at the level of T10. The arms of the forceps were placed within the epidural space on adjacent sides, and their tips touched the floor of the vertebral canal in order to generate reproducible injuries. The spinal cord was then compressed until the spacers connected, and the compression was held for a duration of 5 s. The skin was then joined with biological glue (M3 VetBond). Following the surgery, all mice received analgesia and antibiotics for 3 days post operation (100 μL subcutaneous injection of 1.2 mg/mL Carprieve and Baytril).

### 2.3. PELF-EMF

A SEQEX^®^ device produced by S.I.S.T.E.M.I. S.R.L. (Trento, Italy), certified CSQ ISO-13485, was used for the present study. This device produces complex pulsed electromagnetic fields using an analogue mechanism with a frequency range of 1 to 80 Hz and intensities ranging from 1 to 20 µT. The electromagnetic field produced by the device control unit (on which the electromagnetic field parameters are set) is emitted from a mat containing a Helmholtz coil that generates the pulsed extremely low frequency electromagnetic field (PELF-EMF) 

### 2.4. Treatment Protocol

The mice were treated with PELF-EMF for 7 weeks, 5 times a week, for 4 h. Each treatment consisted of 4 cycles of 1 h duration (Table 1), with treatments starting two hours following compression.

The PELF-EMF protocol included the use of complex modulation using different frequencies. The combinations, referred to as steps, are modulated in order to produce the PEMF for a given time (T-on), followed by a period without emission (T-off) for a blocked time (duration). Each frequency has been chosen for the PELF-EMF protocol based on significant experimental results reported previously:(a)Neuroprotection by the reduction of Glu+-induced excitotoxicity, step 1 (15 Hz) [31,32].(b)Modulation of local inflammation in order to increase the anti-inflammatory effect, steps 2, 3, and 6 (75 Hz and 2 Hz) [25,33].(c)Improvement of the removal of Glu+ from the damaged area (ion cyclotron resonance hypothesis), steps 4 and 5 [34].(d)Stimulation of the recovered area after SCI, step 7 (50 Hz, which also has an anti-inflammatory effect), and neuro-regeneration, steps 8 and 9 (25 Hz and 10 Hz) [25,35,36,37].

### 2.5. Immunohistochemistry 

Cryostat longitudinal sections (20 µm) of fixed frozen tissue were stained using standard immunohistochemistry procedures. Primary antibodies included rabbit anti-GFAP (1:1000, Dako, Copenhagen, Denmark); rabbit anti-Iba1 (1:400; Abcam, Zotal Ltd., Tel Aviv-Yafo, Israel); rat anti-CD68 (1:500; Thermo-Fisher Science, Qiryat Shemona, Israel); rat anti-F4/80 (1:800; Abcam, Zotal Ltd.); mouse anti-βiii tubulin (1:1000; Promega, Kibbutz Beit Ha’Emek, Israel); rabbit anti-BDNF (1:200; Abcam, Zotal Ltd.); rabbit anti-MBP (1:500; Abcam, Zotal Ltd.); and mouse anti-NeuN (1:500; Millipore, Sigma-Aldrich Ltd., Rehovot, Israel). Secondary antibodies included goat anti-rat or rabbit Alexa Fluor 488 and goat anti-mouse Alexa 546 1:1000 (Invitrogen, Sigma-Aldrich Ltd., Rehovot, Israel). Nuclei were visualized using DAPI (Sigma-Aldrich Ltd., Rehovot, Israel)). To obtain the immunofluorescence density of different antibody markers, a series of 20-μm-thick longitudinal sections were cut. For each measurement, sections were taken at 200-μm intervals, and sectioned tissue included both white and grey matter (15 sections per animal for each marker, *n* = 10/group). DAPI immunofluorescence staining was used to define the edge of the lesion and the size was calculated and compared to the vehicle-treated control SCI mice. All measurements were performed using ImageJ software.

### 2.6. Histological Analysis

Longitudinal sections (20 µm; 500-μm intervals, 8 sections per animal; *n* = 10) of fixed, frozen mice spinal cord tissue were stained at 8 weeks post-SCI. They were stained with Cresyl Violet eosin (Sigma-Aldrich Ltd., Rehovot, Israel) for lesion area assessment and Luxol Fast Blue (LFB; Sigma-Aldrich Ltd., Rehovot, Israel) for white matter sparing analysis. The center of each lesion was defined as the section containing the least amount of spared white matter. LFB-positive myelinated areas were measured at the epicentre, and different distances, rostral and caudal from the epicentre, were recorded as specified. 

Sections were imaged by fluorescence microscopy using an Olympus IX83 fluorescence microscope, an ORCA digital camera, and cellSens Dimension Version 3; images were sized using Adobe Photoshop 11 and Illustrator 14. All fluorescence density or intensity measurements were performed using ImageJ software.

### 2.7. Behavioral Analyses

Open field locomotion score: Mice were evaluated for 3 min using the modified Basso–Beattie–Bresnahan (mBBB) 9-point scoring system (control *n* = 10, EMF *n* = 10).

### 2.8. CatWalk

Gait measures were determined using the CatWalk XT 10.6 system5 (Noldus Information Technology, Wageningen, The Netherlands) three times during the experiment: 1 week post injury, 5 weeks post injury, and 7 weeks post injury. It is important to note that we encouraged all mice on the platform to walk at their maximum speed by inflating compressed air on their backs to reduce the variability in running speeds between animals. The visual data were digitized and analyzed using CatWalk XT for static and dynamic gait kinematics, using distance, time, and intensity differences between hind paw prints as measures contributing to gait. Each mouse was placed on the platform and permitted to cross the walking path for at least three compliant runs, when possible (not all mice were cooperative, and some walked less than others), as detected by the CatWalk XT system. Three variables were chosen based on their relevance to human locomotion and human SCI: the base of support (the width, in cm, between the two hind paws), the stride length (the distance, in cm, between subsequent placements of the left hind paw), and the swing speed (speed (distance unit/second) of the paw during swing).

### 2.9. Statistical Analysis 

All statistical analyses were conducted using the GraphPad Prism Program, Version 5.03, for Windows. The significance between the treated and untreated groups was evaluated using the two-tailed *t*-test with 95% confidence when comparing two parameters in the data presented in Figure 1, Figure 2, Figure 3 and Figure 4 (* *p* < 0.05, ** *p* < 0.01, *** *p* < 0.001). The non-parametric Mann–Whitney U test was used to assess the significance of the differences in the CatWalk behavioral analysis (Figure 5; * *p* < 0.05). Data are expressed as mean ± standard error of the mean (SEM), or standard deviation (STDEV), as indicated in the figure legends.

## 3. Results

Since in our previous study, we demonstrated the anti-inflammatory effect of the reduction in glutamate excitotoxicity in spinal cord injured mice, we selected a 15 Hz frequency for use in step 1 [38]. This frequency has shown an anti-glutamatergic effect and a reduction in glutamate levels in vitro [31,32]. In addition, the 2 Hz and 75 Hz frequencies demonstrated a change in local inflammation in the CNS and an increased anti-inflammatory response after spinal cord injury in the rat mode [25,33] and were therefore used in steps 2, 3, and 6 of the protocol. In steps 4 and 5, the 20 Hz and 12 Hz frequencies, respectively, were applied to improve the removal of Glu+ from the damaged area based on the Ion Cyclotron Resonance Hypothesis [34]. The 50 Hz frequency increased the recovery and the anti-inflammatory response after spinal cord injury in rats [25]. We used this frequency in step 7 of the protocol. The 25 Hz and 10 Hz frequencies showed neuroprotective and regenerative effects in the neuronal culture from ischemic brain tissues and promoted the restoration of sensorimotor functions in adult rats with a hemisection of the thoracic spinal cord [36,37]. These frequencies were used in steps 8 and 9.

### 3.1. PELF-EMF Treatment Reduced Astrocyte and Microglia Reactivity

In order to determine whether the PELF-EMF treatment reduced the pro-inflammatory environment two weeks following SCI compression, we examined the degree of activation of microglia/macrophages and astrocytes at the lesion site. Examination of the density of GFAP and Iba1 expression at the lesion site demonstrated that the PELF-EMF treatment significantly decreased glial scarring and inflammation, respectively, at and around the lesion site (Figure 1). Further examination of the M1 phenotype of the activated microglia/macrophages, using Cd68 and F4/80 markers [39,40] was performed. Significantly reduced microglia/macrophage activation was demonstrated by evaluation using CD68+ staining (control 49.9 + 10.9; PELF-EMF 37.2 + 8.4; *** *p* < 0.001) (Figure 2B). Interestingly, some CD68+ cells extended diffusely beyond the lesion cavity borders and glial interface into the grey and white matter in the control groups, compared to the results from the PELF-EMF treated groups. In the treated groups, CD68+ cells were more restricted to the lesion center (Figure 2A). Staining with F4/80, which preferentially stains macrophages and activated microglia, demonstrated positively stained cells mainly at the lesion site and mostly in the control group, with reduced quantified density in the PELF-EMF treated groups (Figure 2C,D).

### 3.2. PELF-EMF Treatment Increases Axonal Survival and BDNF Expression at the Lesion Site

Two months following daily PELF-AMF treatment, significantly higher βIII-tubulin levels were detected in the spinal cord’s white matter at the lesion sites (Figure 3A,B), suggesting that the electromagnetic treatment led to axonal survival at the lesion area. We further examined whether this treatment facilitated pro-survival molecule expression [41]. BDNF, an important neurotrophic factor used in experimental neurotrauma treatments to promote neurogenesis, neuroprotection, axonal sprouting, myelination, and synaptic plasticity, was significantly increased in neuronal cells adjacent to the lesion site in the PELF-EMF group compared to the control group (control 62.7+ 17.3; PELF-EMF 91.6 + 15.1; *** *p* < 0.001) as shown in Figure 3C,D.

### 3.3. PELF-EMF Treatment Moderated the Area of Demyelination at the Lesion Site

BDNF is known to play a role in myelin structure formation, maintenance, and repair, and has been suggested to be a critical factor involved in remyelination and/or structural repair of myelin after neurotrauma [42,43]. Luxol Fast Blue (LFB) histological staining of longitudinal sections at the injury site revealed enhanced myelin sparing in the PELF-EMF-treated mice compared with the untreated mice (Figure 4A,B). Significant myelin loss prevention in the PELF-EMF mice was evident only in the white matter of the lesion center, up to 250 µm rostral and caudal to the injury. Additional analysis of myelin sparing was examined using immunostaining of the myelin binding protein (MBP) in the white matter at the center of the lesion (Figure 4C) and it was found that the MBP density was significantly reduced in the control group when compared to the PELF-EMF-treated group (control 45.2 + 11.7; ELF-EMF 64.16 + 17.08; *** *p* < 0.001). These results may suggest that the electromagnetic treatment reduced the demyelination of the axons through inhibition of the inflammatory response and enhancement of the pro-survival factor secretion.

### 3.4. ELF-EMF Treatment Reduced Functional Deficits a Week after the Injury

As shown in Figure 5, the PELF-EMF-treated mice displayed significant improvement in locomotor recovery at seven days post SCI, but this significance was lost at 14 days after the injury. There was a significant improvement in the mBBB score at 7 days post injury, and the score remained the same up to 5 weeks post-injury (Figure 5A). Mice in the control group reached the same score as the PELF-EMF treated mice at week two, but had significantly lower mBBB scores at one week post-injury.

Similar results were obtained using CatWalk gait analysis. Compared with mice prior to injury, the control group showed significantly lower measurements in swing duration and stride length compare to the PELF-EMF-treated mice, which demonstrated better locomotive abilities (Figure 5B,C). Poorer swing duration measures continued in the control group, even at 7 weeks post-injury. In both groups, the base of support was significantly reduced, compared to that in the pre-operative stage (Figure 5D). These results may suggest that the PELF-EMF treatment produces greater improvement in motor abilities in the first week after injury when compared to control groups, possibly as a result of decreased inflammation. 

## 4. Discussion

Over the past decade, an accumulating body of data suggests that PELF-EMF, in the frequency range of <100 Hz and a field strength < 5 mT, exhibits beneficial therapeutic effects in the treatment of neurotrauma, without any adverse effects. Here we examined its benefit in a mild to moderate SCI model.

In this study, we demonstrated that daily treatment with PELF-EMF, beginning within two hours after SCI, significantly diminished the pro-inflammatory response by reducing microglia and astroglia activation and increasing pro-survival and anti-apoptotic signaling, such as BDNF expression, in mice. Moreover, this treatment exhibited a neuroprotective effect by reducing the axon demyelination in the white matter of the lesion center. The anti-inflammatory and neuroprotective effects of PELF-EMF treatment also improved functional performance during the first week post-injury; however, this improvement was weakened in subsequent weeks.

CNS injury is almost always accompanied by some degree of reactive gliosis, inflammation, and scarring [24,44]. Microglia are the first non-neuronal cells that become activated following neurotrauma, and they are the main source of pro-inflammatory mediators in the CNS. Activated astrocytes and microglia release a wide variety of cytokines, growth factors, and other inflammatory mediators, promoting axonal degeneration, demyelination, and scar formation [45]. Reactive astrocytes densely populate the borders of the injury epicenter, strongly upregulate the intermediate filament protein (such as GFAP) expression, and secrete chondroitin sulfate proteoglycans (CSPGs) into the extracellular matrix, organizing astrocytes into a barrier-like structure that inhibits neural sprouting through this area [44,46,47]. Attenuation of this early inflammatory response to spinal cord injury (SCI) may therefore limit the extent of the secondary tissue injury and, accordingly, the consequent disability [48]. In stroke patients, rehabilitative 4-week treatment with ELF-EMF (F = 40 Hz, Bm = 5 mT), in combination with physiotherapy, reduced both IL-1β plasma and IL-1β mRNA expression levels and increased IL-2 plasma levels without any adverse effects [49]. In addition, no cytotoxic or genotoxic effects were detected in a human mesenchymal stromal cell exposed to an ELF-EMF of 5 Hz, 0.4 mT, for 20 min/day, 3 x/week, for 2 weeks [50].

Here we demonstrated significantly reduced expression of GFAP at and around the lesion site after two weeks of daily treatment with PELF-EMF. Furthermore, the expression of M1 pro-inflammatory CD68+ microglia/macrophages was reduced and limited to the lesion site in the treated group, as compared to the significantly higher and more diffused expression in the untreated group. It is understood that shortly after neurotrauma, as a result of the primary injury, the predominantly M2-like microglia/macrophage environment shifts to an M1 pro-inflammatory type, promoting secondary damage [51,52,53]. It has been shown that the M1 microglia/macrophages indeed have a neurotoxic effect, while the anti-inflammatory M2 type promotes regenerative growth in response to CNS injury [24,52,54]. In our study, long-term neuroprotective markers such as BDNF, which has been identified as a potent promoter of neuronal cell survival and regeneration, were significantly elevated in the lesion site of the treated group two months post-injury. The elevated expression of BDNF, followed by the diminished loss of myelin, supports our hypothesis that the prevention of pro-inflammatory events shortly after spinal cord injury has a long-term neuroprotective effect. Nevertheless, motor function improved significantly in the first week only in the PELF-EMF treated mice, although later, the control group achieved the same status as the treatment group, likely because the injury was mild enough for mice from both groups to gain a high motor recovery score. We believe that the proposed rehabilitative treatment will be less effective in more severe cases of SCI. We could still see a trend of improvement in the gait of the PELF-EMF-treated mice in all the CatWalk parameters we presented in Figure 4 because the injuries to the mice were mild compression injuries, indicating that a combined treatment approach should be examined in the future for better neuro-functional outcomes in SCI patients.

## 5. Conclusions

To date, there is no FDA-approved treatment that can prevent the development of secondary SCI, nor one that induces regenerative processes [5]. Nonetheless, a moderate degree of functional recovery can be achieved using rehabilitative motor training [38,55,56]. Rehabilitation is probably one of the most important interventions following neurotrauma, and it can potentially improve a patient’s quality of life significantly. However, multi-trauma and the management of other medical complications in the acute post-injury setting often preclude or complicate early rehabilitation that demands the patient’s active participation. Therefore, a simple, safe, and approved-for-use rehabilitation device that can be combined with traditional therapy, without the need to involve the patient in active participation immediately after a spinal cord injury, may have significant therapeutic value.

## Figures and Tables

**Figure 1 biomedicines-10-00325-f001:**
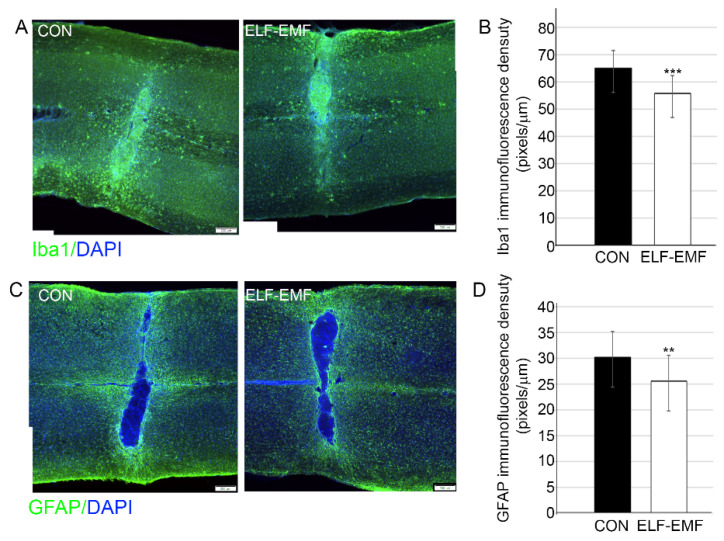
Decreased microglia activation and astrogliosis in PELF-EMF-treated mice after SCI. Two weeks after SCI (**A**) Representative images of the lesion site of Iba1 (green) immunostaining; DAPI (blue) demonstrate the lesion site. The scale bar is 200 μm. (**B**) Quantitation of Iba1 at the lesion site shows a significant decrease in the PELF-EMF-treated compared to the non-treated mice (*n* = 8 in each group; *** *p* < 0.001). (**C**) Representative images of the lesion site of GFAP (green) immunostaining; DAPI (blue) demonstrate the lesion site. The scale bar is 200 μm. (**D**) Quantitation of GFAP at the lesion site shows a significant decrease in the PELF-EMF-treated compared to the non-treated mice (*n* = 8 in each group; ** *p* < 0.01).

**Figure 2 biomedicines-10-00325-f002:**
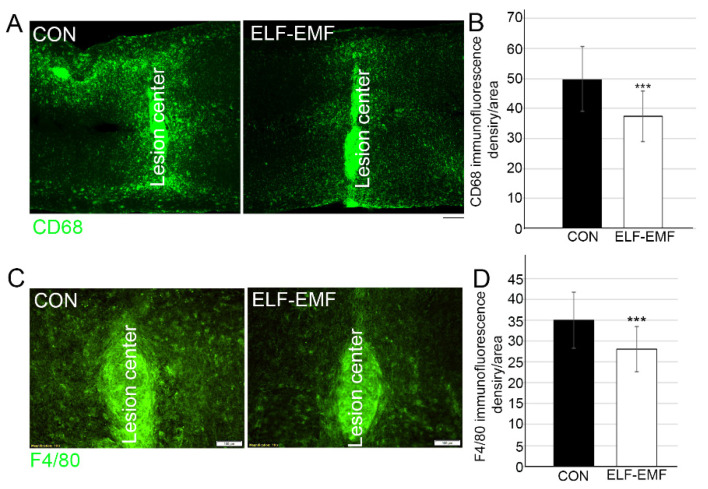
Decrease of other microglia activation markers in PELF-EMF-treated mice after SCI. Two weeks after SCI (**A**) Representative images of the lesion site of CD68 (green) immunostaining; DAPI (blue) demonstrate the lesion site. The scale bar is 100 μm. (**B**) Quantitation of CD68 at the lesion site shows a significant decrease in the PELF-EMF-treated compared to the non-treated mice. The results are mean ± SD (*n* = 8/group; *** *p* < 0.001). (**C**) Representative images of the lesion site of F4/80 (green) immunostaining; DAPI (blue) demonstrate the lesion site. The scale bar is 100 μm. (**D**) Quantitation of F4/80 at the lesion site shows a significant decrease in the PELF-EMF-treated compared to the non-treated mice. The results are mean ± SD (*n* = 8 in each group; *** *p* < 0.001).

**Figure 3 biomedicines-10-00325-f003:**
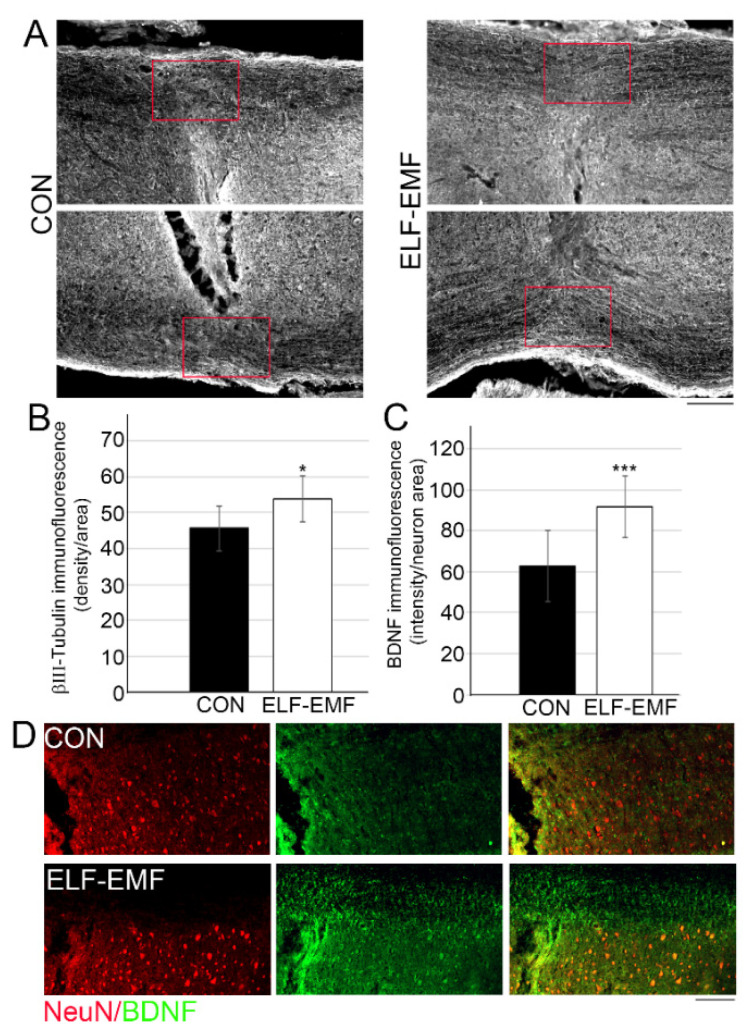
The increased axonal survival and BDNF expression in PELF-EMF-treated mice after SCI. Two months after SCI (**A**) Representative images of the white matter at the lesion site of the βIII-tubulin immunostaining; the scale bar is 100 μm. (**B**) Quantitation of the βIII-tubulin density immunostaining at the lesion site shows a significant increase in the PELF-EMF-treated compared to the non-treated mice. The results are mean ± SD (the red box represents the area of analysis; *n* = 10 in each group; * *p* < 0.05). (**C**) Quantitation of the BDNF (green) expression density in neuronal cells (NeuN in red) close to the lesion site shows a significant increase in the PELF-EMF-treated compared to the non-treated mice. The results are mean ± SD (*n* = 10 in each group; *** *p* < 0.001). (**D**) Representative images of lesion site of the BDNF (green) and NeuN (red) immunostaining; DAPI (blue) demonstrate the lesion site. The scale bar is 100 μm.

**Figure 4 biomedicines-10-00325-f004:**
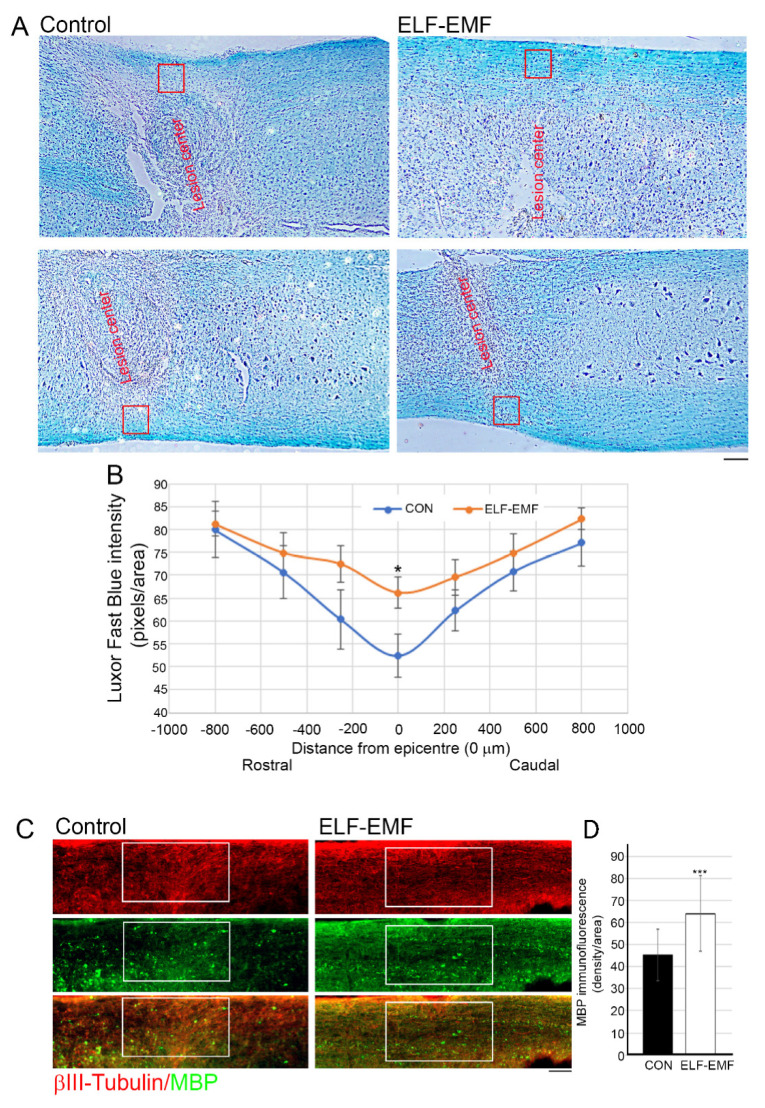
The reduced axonal demyelination and increased MBP expression in PELF-EMF-treated mice after SCI. Two months after SCI (**A**) Representative images both sides of the white matter of the spinal cord stained with LFB. The injury epicentre (0) is marked in the red box. The scale bar is 100 μm. (**B**) Quantitative analysis of residual myelin in different rostral and caudal distances from the epicentre of the white matter in the PELF-EMF and the control treated groups shows a significant decrease in demyelination in the PELF-EMF-treated mice. Data represent the mean ± SD (*n* = 10/group; * *p* < 0.05). (**C**) Representative images of the white matter at the lesion site with MBP immunostaining; the scale bar is 100 μm. (**D**) Quantitation of the MBP density immunostaining at the lesion site shows a significant increase in the PELF-EMF-treated compared to the non-treated mice. Results are mean ± SD (*n* = 10/group; *** *p* < 0.001).

**Figure 5 biomedicines-10-00325-f005:**
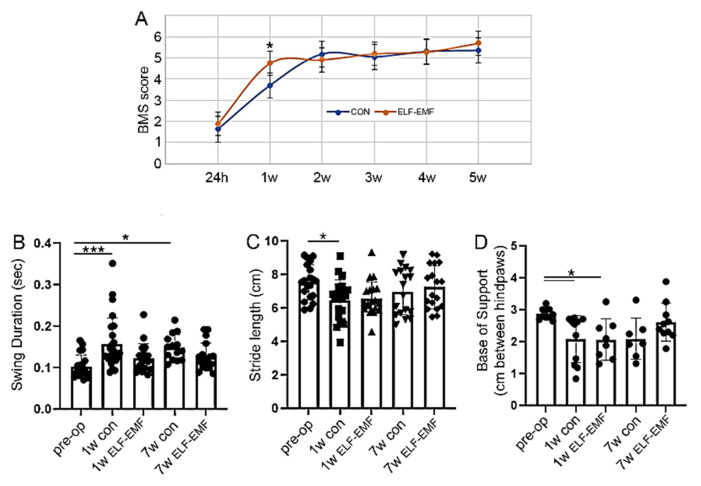
The improved motor function in the PELF-EMF-treated mice after SCI. Two months after SCI, the motor function recovery of the mice was assessed using different behavioral tests. (**A**) mBBB scores demonstrated a significant improvement in the first week after SCI in the PELF-EMF-treated mice compared to the control; the subsequent weeks showed no difference between the groups (*n* = 10 animals/group. The results are the mean ± SEM (* *p* < 0.05; non-parametric Mann–Whitney test, α set to 5%). The CatWalk analysis of swing duration (**B**), stride length (**C**), and the base of support (**D**) (*n* = 10 animals/group); the data are expressed as the mean ± SEM, one-way ANOVA followed by Bonferroni’s multiple comparison test; * *p* < 0.05, *** *p* < 0.001).

**Table 1 biomedicines-10-00325-t001:** The PELF-EMF 9-cycle treatment protocol.

Step	Frequency (Hz)	Field Intensity (µT)	T-On (min)	T-Off (min)	Duration (min)
1	15	20	2	3	14
2	75	20	4	1	12
3	2	20	2	3	14
4	20	20	3	1	14
5	12	20	3	1	14
6	3	20	3	1	14
7	50	20	2	1	12
8	25	20	5	2	12
9	10	20	5	1	14

## Data Availability

The data presented in this study are available on request from the corresponding author.

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
