# Peer review of "Treatment with Pulsed Extremely Low Frequency Electromagnetic Field (PELF-EMF) Exhibit Anti-Inflammatory and Neuroprotective Effect in Compression Spinal Cord Injury Model"

_biomedicines, 2022, doi:10.3390/biomedicines10020325_

Round 1

Reviewer 1 Report

In the manuscript entitled: “Treatment with pulsed extremely low frequency electromagnetic field (PELF-EMF) exhibit anti-inflammatory and neuro- 3 protective effect in compression spinal cord injury model” by Goldshmit, Y. et al.

The authors describe the effects of PELF-EMF as a neuroprotective treatment for SCI. The authors found that this therapeutically approach significant diminished inflammatory cell infiltration and astrocyte activation reducing Iba1, CD68 + cells and GAFP at the lesion borders, and increased pro-survival signaling, such as BDNF on neuronal cells. Despite the outcomes showed in the present work, I have a major concern regarding the conclusion that the authors arrived based on the potential use of this technique to the SCI treatment, since the authors show that there was not a  significant difference in the locomotor recovery as compared to controls. Therefore the locomotor recovery outcomes triggered using this rehabilitation device could not significant overcome the motor deficits caused by SCI.

Major comments

1- The authors need to improve the PELF-EMF treatment (For example modifying Frequency, Intensity, Duration) in order to attempt obtain a better locomotor recovery outcome if the aim of the authors is looking to apply this therapeutically approach in human.

2- Figures need to be improved. High quality magnification images should be added in order to show with more detail the shape of axons, myelin, astrocytes and microglia.

3- Neuronal reconnection experiments after contusion (CON vs ELF-EMF) should be done.

Reviewer 2 Report

This is an interesting manuscript, but does suffer from several methodological limitations.

1) Although a 2 month old mouse is considered "adult", this is still a very "young" adult and that could affect the results.  This is not commented upon.

2) A second factor which could affect the results is that the compression is quite mild.  This is, however, commented upon.

3) Three minute BBB evaluation is quite short for thorough analysis.

4) Blowing on the mouse's back is a rather unscientific and variable method which could impact the results.  Moreover, its impossible to tell which animals had this technique vs the standard compressed air stimulus.

5) As far as I can tell, the statistical analysis does not use repeated measures when appropriate.  Furthermore, it is impossible to discern when Bonferroni was used and when it wasn't.  Overall, the statistics are very poorly described.

6) In that the behavioral results are only different between the experimental and control groups at week 1: a) what is the relevance of the immunostaining at 2 weeks, and b) It is not reasonable to make the blanket statement that the treatment improved the locomotor activity.

7) Minimal validation is given for the specific frequencies used  at the specific timepoints and durations listed.

Round 2

Reviewer 2 Report

I am satisfied with the authors revisions.